# Pre-Bleaching Coral Microbiome Is Enriched in Beneficial Taxa and Functions

**DOI:** 10.3390/microorganisms12051005

**Published:** 2024-05-16

**Authors:** Laís F. O. Lima, Amanda T. Alker, Megan M. Morris, Robert A. Edwards, Samantha J. de Putron, Elizabeth A. Dinsdale

**Affiliations:** 1Marine Biology, Scripps Institute of Oceanography, University of California San Diego, La Jolla, CA 92093, USA; lfolima@ucsd.edu; 2San Diego State University, San Diego, CA 92182, USA; 3Innovative Genomics Institute, University of California, Berkeley, SA 5045, USA; aalker@sdsu.edu; 4Lawrence Livermore National Laboratory, Livermore, CA 94550, USA; m.morris2187@gmail.com; 5Flinders Accelerator Microbiome Exploration, Flinders University, Bedford Park, SA 5042, Australia; robert.edwards@flinders.edu.au; 6Bermuda Institute of Ocean Sciences, St. George’s GE01, Bermuda; s.deputron@bios.asu.edu

**Keywords:** holobiont, thermal tolerance, nutrient cycling, coral reefs

## Abstract

Coral reef health is tightly connected to the coral holobiont, which is the association between the coral animal and a diverse microbiome functioning as a unit. The coral holobiont depends on key services such as nitrogen and sulfur cycling mediated by the associated bacteria. However, these microbial services may be impaired in response to environmental changes, such as thermal stress. A perturbed microbiome may lead to coral bleaching and disease outbreaks, which have caused an unprecedented loss in coral cover worldwide, particularly correlated to a warming ocean. The response mechanisms of the coral holobiont under high temperatures are not completely understood, but the associated microbial community is a potential source of acquired heat-tolerance. Here we investigate the effects of increased temperature on the taxonomic and functional profiles of coral surface mucous layer (SML) microbiomes in relationship to coral–algal physiology. We used shotgun metagenomics in an experimental setting to understand the dynamics of microbial taxa and genes in the SML microbiome of the coral *Pseudodiploria strigosa* under heat treatment. The metagenomes of corals exposed to heat showed high similarity at the level of bacterial genera and functional genes related to nitrogen and sulfur metabolism and stress response. The coral SML microbiome responded to heat with an increase in the relative abundance of taxa with probiotic potential, and functional genes for nitrogen and sulfur acquisition. Coral–algal physiology significantly explained the variation in the microbiome at taxonomic and functional levels. These consistent and specific microbial taxa and gene functions that significantly increased in proportional abundance in corals exposed to heat are potentially beneficial to coral health and thermal resistance.

## 1. Introduction

Microbial symbioses are the engine of coral reef ecosystems. Corals associate with endosymbiotic dinoflagellates of the family *Symbiodiniaceae* and a diverse microbiome (e.g., bacteria, archaea, viruses), are functioning as a unit and forming a holobiont [1]. The coral holobiont depends on key services such as nitrogen and sulfur cycling mediated by the associated microbiome [2,3,4,5]. The coral surface mucous layer (SML) sustains a diverse and abundant community of these microbial partners [6,7,8,9,10,11]. The coral microbiome benefits from the high nitrogen content and organic matter in the SML [5,12] and provides protection against coral pathogens via the production of antimicrobials [13,14]. However, coral-associated microbial communities are sensitive to environmental changes, especially to increased temperature, which disrupt the beneficial services provided to the holobiont [15,16,17,18].

Coral reefs are at great risk of collapse as coral bleaching (i.e., loss of algal symbionts) and disease outbreaks have become more frequent in the last two decades, particularly correlated to rising seawater temperature, leading to major losses in coral cover worldwide [19,20,21,22,23]. These losses are pronounced in the shallow water reefs of the Caribbean, where an overall decline in coral cover of up to 59% has occurred since 1984 [24]. Corals live at their upper thermal limits, and therefore thermal thresholds may not be able to adjust to projected rises in seawater temperature in times of rapid environmental change [25,26,27,28].

The mechanisms of resistance to environmental change in the coral holobiont are not completely understood, but the associated microbial community is a potential source of acquired heat-tolerance [29,30,31,32,33]. Beneficial bacteria isolated from coral tissue and surrounding water containing genes involved in nitrogen (e.g., nitrogenase *nifH*, nitrification *nirK*) and sulfur cycling (e.g., DMSP degradation *dmdA*) ameliorated the effects of increased temperatures (from 26 °C to 30 °C) and opportunistic pathogens on *Pocillopora* [31] and *Mussismilia hispida* [32] corals in aquarium manipulations. Microbiome transplants extracted from heat-tolerant corals mitigated the bleaching response of *Pocillopora* and *Porites* corals exposed to a short-term heat treatment (from an ambient 29 °C to a 34 °C heat for 9 h) [33]. There is also a potential evolutionary role of the microbiome as a source of genes (e.g., stress response genes) that can be used in thermal resilience and disease protection via mechanisms such as horizontal gene transfer [34]. The use of metagenomics associated with physiological data in experimental settings is a recommended approach to further explore the role of the coral microbiome in heat tolerance and stress response [35]. Therefore, changes in the proportions of coral–microbial taxa and gene functions in response to heat could reveal key players in the microbiome that may be providing beneficial services to support a thermally challenged holobiont.

Here, we investigate the effects of increased temperature on the taxonomic and functional profiles of coral SML microbiomes. We addressed these aims by exposing corals to a pre-bleaching heat treatment and analyzing coral–algal physiological parameters and microbial taxa (genus level) and gene functions (nitrogen metabolism, sulfur metabolism, and stress response) associated with the coral SML using shotgun metagenomics. As the coral SML is in intimate contact with the external environment, acting as a conduit between the water column and host physiology, we hypothesize that changes in coral host physiological parameters are correlated to the dynamics of the SML microbiome.

## 2. Materials and Methods

### 2.1. Field Sampling

We selected *Pseudodiploria strigosa* (Dana, 1846) as the coral host species because it is widely distributed in the Caribbean and, more specifically, across the Bermuda platform in the sampling area of this study. In Bermuda, a high-latitude subtropical coral reef system, corals are exposed to a wide annual range in temperature fluctuations and reef zone-specific thermal regimes ranging from more fluctuating profiles in the inner lagoon patch reefs (temperature differences of ~13–15 °C between winter averages of 16–17 °C and summer averages of 30–31 °C) and milder conditions in the outer reefs (temperature differences of ~10 °C between seasonal averages of 19 and 29 °C) [11,36,37,38,39]. The sampling period occurred between 18 May and 22 May 2017, late spring in the northern hemisphere, when environmental conditions between the two reef zones, especially temperature, are similar (~23–25 °C) [10]. The mucus from *P. strigosa* was collected from two colonies from the inner and outer reef zones (*n* = four colonies in total) to describe the natural spatial variability in the coral microbiome [10,11]. After the mucus sampling, each coral colony was collected using a hammer and chisel to be used in the experiments to assess heat resistance. The colonies (*n* = four in total, two from each reef zone) were carefully placed in a cooler with seawater and transported to the Bermuda Institute of Ocean Sciences. The collections were performed via SCUBA diving at 4–6 m depth.

### 2.2. Colony Preparation and Acclimation

After collection, the coral colonies were placed in an outdoor tank with an opaque roof supplied with unfiltered flowing seawater for a week to acclimatize. Each colony was fragmented into four pieces using a tile saw and then tagged, and all non-coral tissue areas were cleaned and covered in reef-safe epoxy. The colony fragments, here referred to as “coral nubbins”, were left to acclimatize in the indoor system for two weeks (ambient temperature 25 °C ± 1 °C). Using coral nubbins in laboratorial experiments is a way to expose the same colony and microbiome to different treatments, allowing for the assessment of genetic variability among treatments, as well as reducing the number of coral colonies collected. The nubbins were monitored and only healthy ones, assessed by the recovery of the coring process (outward growth as opposed to tissue regression) and visual endosymbiont function, as measured by color and fluorescence readings (Fv/Fm > 0.6) with a pulse amplitude-modulated fluorometer (DIVING-PAM Walz Inc., Effeltrich, Germany), were used in the experiments.

### 2.3. Temperature Treatments

After the acclimation period, eight coral nubbins (A.1, A.2, B.1, B.2, C.1, C.2, D.1, D.2) from four distinct colonies of coral *P. strigosa* (colonies A and B, from the inner reef zone and colonies C and D from the outer reef zone) were exposed to a high temperature treatment (29 °C ± 1 °C, nubbins A.1, B.1, C.1, D.1), while the remaining were maintained at ambient conditions (control, 25 °C ± 1 °C; nubbins A.2, B.2, C.2, D.2). The temperature was gradually increased from ambient conditions to hot conditions over two days then maintained within the target range. An increase of 4–5 °C from ambient temperatures is commonly used in coral microbiome heat treatments (31–33). We assumed that the temperature increase from spring (25 °C) to summer averages (29 °C) in a week would suffice to trigger a heat response, but not bleaching, in the coral holobiont. The indoor temperature-controlled tank system consisted of two fiberglass trays fed from a common header tank of filtered seawater. Each tray table was temperature-controlled by aquarium chillers and heaters. Light intensity was controlled by aquarium lights (Storm and Storm X LED Controllers, CoraLux LLC) in a 12:12 h photoperiod. The temperature was measured every 5 min throughout the experiment by pairs of HOBO data loggers (Onset Corp., Marleston, Australia) and placed in each tray table, and the light intensity was measured by a LiCor PAR sensor, Lincoln, Nebraska, USA.

### 2.4. Physiological Measurements

Productivity and respiration rates were measured using dissolved oxygen probes in customized transparent glass chambers (approx. 4 L) to detect oxygen evolution (production), and oxygen uptake (respiration rate). Each chamber had an oxygen sensor spot (PyroScience, Aachen, Germany) glued to the inner chamber wall and a small in-line aquarium pump for water mixing during incubation. Oxygen concentration was monitored using a fiber-optic sensor (PyroScience). Colonies were set individually inside the chamber that was bathed in and filled with the filtered, temperature-controlled seawater from the experimental tray table. Dissolved oxygen concentration was measured in the chamber during a 1 h dark period to calculate dark respiration rates (R) and, subsequently, during a light exposure period of 1 h to calculate net productivity (NP). The difference between the dissolved oxygen concentration at incubation, i.e., the start, and at the end of the experiment was normalized over time and colony surface area [40]. Surface area was determined by the aluminum foil technique, carefully following the contours of the coral surface [41], and image analysis (Image J (https://imagej.net/), US National Institutes of Health, Washington, DC, USA) of planar digital images. The dark respiration and photosynthesis rates were analyzed according to their linearity over time for quality control. Gross productivity (GP) was calculated (gross productivity = net productivity + dark respiration). Photosynthesis to respiration (GP:R ratios) were calculated as a ratio of gross productivity (GP) to respiration (R) to estimate whether production by the *Symbiodiniaceae* cells exceeds the maintenance requirements of both the symbiont and the coral host [42]. The maximum photochemical efficiency of photosystem II (i.e., effective quantum yield) was quantified via pulse–amplitude modulate (PAM) fluorometry [43,44]. After an overnight dark acclimation period, fluorescence (Fm and F0) was measured by saturation pulses at three random spots on each colony to calculate the colony’s average Fv/Fm. The maximum photochemical efficiency of the PSII (Fv/Fm, Fv = Fm − F0) was calculated based on fluorescence measurements using a pulse amplitude modulated fluorometer (DIVING-PAM Walz Inc., Effeltrich, Gremany).

### 2.5. Experiment Design

Productivity and respiration rates and photochemical efficiency were measured the day before the heat treatment started (post-acclimation or pre-treatment) and on day 7, which is the final day of the experiment (post-treatment). The coral mucus for metagenomic analysis was collected at the end of each incubation using a two-way 50 mL syringe that is filled with 0.02 µm of filtered seawater [10,11]. The filtered seawater is flowed across the coral surface, dislodging the mucus and associated microbes, which are then sucked up via the recirculating tube, and the resulting sample is pushed through a 0.22 µm sterivex for DNA extraction. Metagenomes were sequenced from coral nubbins (*n* = 8), representing four different coral colonies (*n* = 2 colonies per reef zone) that had their mucus sampled before and after the experimental treatments (*n* = 16 metagenomes total). These metagenomes were compared with the ones from the same colonies collected in situ [10].

### 2.6. Metagenomic Analysis

Microbial DNA from the coral mucus collected on the 0.22 µm Sterivex was extracted using a modified Macherey–Nagel (Dűren, Germany) protocol using a NucleoSpin column for purification. DNA was stored at −20 °C until quantification with Qubit (Thermo Fisher Scientific, Waltham, MA, USA) [45]. The Swift kit 2S plus (Swift Biosciences, Ann Arbor, Michigan, USA) was used for library preparation since it provides good results from small amounts of input DNA, which is characteristic of microbial samples collected from the surface of the host [46,47]. All samples were sequenced by the Dinsdale lab on Illumina MiSeq ( San Diego, CA, USA) at San Diego State University. The sequenced DNA was analyzed for quality control using PrinSeq V1 [48] before annotation to remove and trim low-quality sequences (i.e., exact duplicates, sequences that contained N’s, and sequences that had a *Q*-score < 20). The metagenomes were annotated through MG-RAST [49], applying the minimum cutoff parameters of 1 × 10^−5^ *e*-value, 70% identity, and an alignment length of 30 nucleotides, using the RefSeq database (Pruitt et al., 2007) [50] for taxonomic annotations and the SEED database (Overbeek et al., 2014) [51] for functional annotations.

We used metagenomics to describe the abundance of bacteria functions that are potentially experiencing strong selective pressure in response to environmental conditions [52,53,54]. Although metagenomics does not measure which functional genes are being expressed at the point the sample was taken (i.e., metatranscriptomics), it shows which microbial functional genes are over- or underrepresented across microbiomes [55,56].

For the taxonomic composition, the metagenomes were filtered in MG-RAST to include only bacteria at the level of genera. For the functional composition, the metagenomes were filtered in MG-RAST for stress response, nitrogen metabolism, and sulfur metabolism. We selected these three broad functional gene groups (SEED subsystem level 1) because they have the greatest relevance for corals under heat stress [2,3,17,57,58,59,60,61,62]. The number of sequence hits for each microbial taxon or function is represented as the relative abundance by calculating the proportion of sequence hits for that parameter over the total number of sequences annotated for that metagenome. Metagenomes were compared using proportional abundance, which is preferred to rarefaction [63,64,65]. Bacteria accounted for approximately 99% of the annotation; therefore, we are only analyzing bacterial taxa and gene functions in this study.

### 2.7. Statistical Analysis

Statistical analyses were conducted using PRIMER-e version 7 plus PERMANOVA (PRIMER-E Ltd., Plymouth, UK) and Statistical Analyses of Metagenomic profiles (STAMP) [66] software V2.1.3. Using PRIMER-e for multivariate analysis, the relative abundances in the metagenomes annotated at the level of bacterial genus, nitrogen metabolism, sulfur metabolism, and stress response were square root-transformed and resemblance matrices were generated based on Bray–Curtis similarities. The coral physiology data (GP:R ratios and Fv/Fm) was log (x + 1)-transformed, normalized, and resemblance matrices were generated based on Euclidean distances. Significant differences in the metagenomic and physiological data across the temperature treatments (pre-treatment, ambient, and heat) were identified by the permutational multivariate analysis of variance (PERMANOVA). Distance-based linear models (DistLM) using the Akaike information criterion corrected for small sample sizes (AICc) were applied to test the coral physiology data as predictor variables to the relative abundances of bacterial taxa, nitrogen metabolism genes, sulfur metabolism genes, and stress response genes. These relationships were visualized as boxplots, principal coordinate analyses (PCO), and distance-based redundancy analyses (dbRDA). We also used PRIMER-e to calculate Pielou’s evenness index (J′) and Shannon’s diversity index (H′) of microbial genera. Multiple comparisons of either taxa or functions across the temperature treatments were conducted in STAMP using ANOVA/Tukey–Kramer, Welch’s pairwise comparisons, and Benjamini–Hochberg FDR corrections.

## 3. Results

### 3.1. Coral–Algal Physiology

Corals showed a significant reduction in physiological performance (GP:R ratios and Fv/Fm as multivariate response variables) when exposed to the heat treatment (pairwise PERMANOVA, pre-treatment × heat: t = 4.61, P(perm) = 0.004; ambient × heat: t = 4.48, P(perm) = 0.03). Pre-treatment (*n* = 8, T = 25 °C, day 1 of the experiment) mean values of Fv/Fm (0.64 ± 0.016) and GP:R (2.1 ± 0.37) remained relatively stable in the ambient treatment (*n* = 4, T = 25 °C, day 7; Fv/Fm = 0.62 ± 0.012, GP:R = 2.1 ± 0.15) (Figure 1), while declining under heat (*n* = 4, T = 29 °C, day 7; Fv/Fm = 0.58 ± 0.017, GP:R = 1.44 ± 0.17) (Figure 1). There were no visual signs of the paling or bleaching of coral nubbins during the experiment, thus enabling an investigation into how the microbiome changed before bleaching became apparent.

### 3.2. Coral SML Microbiome

The metagenomes associated with the coral SML of *P. strigosa* were sequenced at high coverage, ranging from 356,426 to 1,296,198 sequence counts (Appendix A). Of a total of 595 taxa, 34 were unclassified at the genus level and maintained in the analysis according to the highest possible taxonomic annotation. Microbial richness (S, number of genera) ranged from 578 to 587 genera and did not significantly change between microbiome samples collected in situ and across the experimental treatments (Appendix A). Diversity (H′) was highest in situ due to a decrease in evenness (J′) under experimental conditions. Coral microbiomes at the level of bacterial genera retained a 75% average Bray–Curtis similarity under experimental conditions when compared to in situ conditions (before colony collection) (Appendix A).

The principal coordinate analyses of bacterial genera and genes related to nitrogen metabolism, sulfur metabolism, and stress response (Figure 2), showed that dispersion (i.e., dissimilarity, β-diversity) was lower among metagenomes from heat-exposed corals than from pre-treatment and control groups. Average similarities (Bray–Curtis) within treatments at the level of bacterial genera (pre-treatment: 88.69%; ambient: 90.09%; heat: 95.07%), nitrogen metabolism genes (pre-treatment: 93.67%; ambient: 93.75%; heat: 97.22%), sulfur metabolism genes (pre-treatment: 92.99%; ambient: 93.91%; heat: 96.75%), and stress response genes (pre-treatment: 96.41%; ambient: 97.28%; heat: 98.41%) increased after heat exposure. Pairwise PERMANOVAs concluded that the relative abundances of microbial genera (t = 2.6, P(perm) < 0.03), nitrogen metabolism genes (t = 2.52, P(perm) = 0.03), and sulfur metabolism genes (t = 3.14, P(perm) = 0.03) were significantly different between pre-treatment and heat-exposed microbiomes. Metagenomes from corals at ambient conditions did not significantly change when compared to pre-experiment or heat treatments. The relative abundances of microbial stress response genes showed no significant change across treatments, suggesting a pre-bleaching condition.

Heat exposure led to a significant increase in the relative abundances of *Ruegeria* (t = −2.38, corrected *p*-value < 0.02), *Roseobacter* (t = −2.15, corrected *p*-value < 0.001), *Oceanibulbus* (t = −1.58, corrected *p*-value < 0.03), *Chromohalobacter* (t = −0.84, corrected *p*-value < 0.02), and *Halomonas* (t = −0.87, corrected *p*-value < 0.02), according to Welch’s pairwise comparisons among the top 20 most abundant taxa in the coral microbiome (Figure 3). In contrast, there was a significant decrease in the relative abundances of *Shewanella* (t = 0.82, corrected *p*-value < 0.001), *Synechococcus* (t = 0.88, corrected *p*-value < 0.04), and *Vibrio* (t = 0.284, *p*-value < 0.02) in the microbiome of corals exposed to heat treatment (Figure 3).

Stress response genes were investigated across the treatments; however, there was no significant difference in the proportional abundance of these genes (Figure 4A). Within nitrogen metabolism, the relative abundances of microbial gene pathways related to amidase with urea and nitrile hydratase (t = −0.363, corrected *p*-value < 0.0001), allantoin utilization (t = −1.079, corrected *p*-value = 0.042), and nitrogen fixation (t = −0.130, corrected *p*-value = 0.049) increased in the coral microbiome under heat exposure, while nitrosative stress (t = 1.628, corrected *p*-value = 0.033), and ammonia assimilation (t = 4.51, corrected *p*-value = 0.037) decreased (Figure 4B).

Sulfur metabolism microbial genes also changed in relative abundance after heat exposure of corals. There was an increase in glutathione utilization (t = −1.70, corrected *p*-value < 0.0001), sulfur oxidation (t = −8.08, corrected *p*-value < 0.0001), and taurine utilization (t = −1.44, corrected *p*-value < 0.0001) and a decrease in inorganic sulfur assimilation (t = 8.04, corrected *p*-value < 0.0001) (Figure 4C).

### 3.3. Coral Microbiome in Relationship to Coral–Algal Physiology

Distance-based redundancy analysis (dbRDA) showed the influence of the coral physiological performance on the coral microbiome structure at genus level, where heat-exposed metagenomes cluster away from the control treatments that are more positively correlated to GP:R ratios and Fv/Fm (Figure 5). Marginal tests for distance-based linear models (DistLM) indicated that Fv/Fm significantly explained the observed compositional variation of bacterial genera (Pseudo-F = 4.2, *p* = 0.007, AICc = 71.2, R2 = 0.25) and of genes related to nitrogen (Pseudo-F = 4.5, *p* = 0.004, AICc = 51.9, R2 = 0.26) and sulfur metabolisms (Pseudo-F = 6.3, *p* = 0.01, AICc = 56.7, R2 = 0.32). The relative abundances of bacterial stress response genes could not be significantly explained by coral’s physiological variables (AICc = 32.0, R2 = 0.10).

## 4. Discussion

### 4.1. The Coral SML Microbiome Response to Heat Exposure

Heat exposure affected coral–algal physiological performance and caused significant compositional and functional changes in the microbiome of *P. strigosa*. Dissimilarity (i.e., β-diversity) decreased in heat-exposed microbiomes at the level of bacterial genera, stress response, nitrogen metabolism, and sulfur metabolism genes. According to the Anna Karenina principle, taxonomic β-diversity is expected to increase among coral microbiomes under stress and in a dysbiotic state [67,68]. However, dissimilarity among coral microbiomes exposed to hyper salinity [69], bleaching conditions [70], and higher temperatures (this study, Figure 2A) did not increase, indicating that some coral–microbial systems show a specific and consistent response to environmental changes. A similar pattern was observed among the coral colonies used in this study in their natural environment (i.e., prior to collection and experimental manipulation). Dissimilarity was lower among coral SML microbiomes that are naturally more exposed to temperature changes in the inner reefs and higher among those under a milder temperature regime in the outer reefs of Bermuda [11].

As expected, the coral SML microbiome structure changed after coral colonies were removed from their native environment and fragmented. Experimental settings and manipulation inevitably affect the coral microbiome [71,72]; therefore, the results of this experiment may not entirely reflect the microbiome dynamics that occur in the natural environment. However, the experimental metagenomes retained a significant proportion of the taxonomic profile seen in the samples collected in situ (75% average similarity).

The relative abundance of specific bacterial genera (Figure 3) and nitrogen and sulfur gene functions (Figure 4) changed consistently across coral replicates that were exposed to the heat treatment, while in the ambient treatment these changes were more variable and not significantly different from pre-treatment coral microbiomes. Diversity (H’) remained high both in ambient and heat conditions. Here we present a case for a coral SML microbiome response to increased temperatures that is potentially beneficial to the holobiont at taxonomic and functional levels.

### 4.2. Bacterial Taxa

*Ruegeria* and *Roseobacter* significantly increased in relative abundance in response to heat stress (Figure 2). High abundances of *Rhodobacterales* such as *Ruegeria* and *Roseobacter* in juvenile and adult corals suggest they play a key role in coral fitness [71,73], nitrogen acquisition and remineralization [74,75], and sulfur cycling [2]. *Ruegeria* are among the three genera that are most frequently associated with coral species [76]. Some strains of *Ruegeria* have probiotic potential as they inhibit the growth of the pathogen *Vibrio coralliilyticus* [77] and support corals in withstanding heat stress [31,32,78]. In fact, *Vibrio* spp. decreased in relative abundance in the coral microbiome under heat treatment, indicating that *P. strigosa* was able to keep these potential pathogens in check.

Another beneficial service that the microbiome could be providing to the coral holobiont is the supplementation of energy and nutrients via heterotrophic feeding. The algal symbionts *Symbiodiniaceae* provide 75–100% of the daily metabolic requirements of the coral holobiont via photosynthesis [79,80]. However, high temperatures disrupt the coral–algal physiology, leading to insufficient energy intake to support the coral holobiont [80,81]. In addition to photosynthates provided by *Symbiodiniaceae*, corals can obtain fixed carbon from the ingestion of bacterioplankton trapped in the coral SML [74,79,80]. Corals preferentially feed on cyanobacteria *Synechococcus*, especially when recovering from heat stress and bleaching [74,82,83]. We confirm lower relative abundances of *Synechococcus* in the coral SML after heat exposure, which is an indication that the corals were hypothetically consuming these microbes to acquire nutrients and compensate for lower productivity of the algal symbiont.

### 4.3. Nitrogen Metabolism

The SML microbiomes of *P. strigosa* showed that bacterial genes related to nitrogen and sulfur metabolism responded to heat exposure. Ammonia assimilation genes decreased in relative abundance after heat exposure, indicating that less ammonia is being used in the biosynthesis of bacterial compounds by the microbiome (Figure 4B). Coral-associated bacteria compete with *Symbiodiniaceae* for host-generated ammonia, which is the preferred form of nitrogen of the symbiotic algae [5,35]. Lower ammonia assimilation by the coral SML microbiome under heat exposure is beneficial to the holobiont because more ammonia may be available to the algal symbiont to support photosynthesis and prevent coral bleaching.

Nitrosative stress genes allow bacteria to detoxify nitric oxide (NO) and reactive nitrogen species (RNS) involved in the denitrification process [84]. Nitrosative stress genes decreased in relative abundance under elevated heat, while denitrification and NO synthase genes did not change in proportion (Figure 4B). Therefore, the lower relative abundance of nitrosative stress genes could be a result of stability in the levels of NO and RNS in the coral SML microbiome.

Nitrogen incorporation genes, via allantoin utilization and nitrogen fixation, significantly increased in the coral SML with elevated heat (Figure 4B). Allantoin is a urea-related compound produced by plants that is a nitrogen source for bacteria and the method of transporting fixed nitrogen to plants when nitrogen is limited [85,86]. Bacteria provide about 11% of the nitrogen required by *Symbiodiniaceae* via nitrogen fixation in the coral holobiont [35,57]. The significant increase in the proportions of allantoin utilization and nitrogen fixation genes is a key beneficial service offered by the coral SML microbiome to supply nitrogen to the holobiont to withstand increasing temperatures.

### 4.4. Sulfur Metabolism

The relative abundances of bacterial genes related to glutathione utilization, sulfur oxidation, and taurine oxidation increased with heat exposure in the coral SML microbiome. Glutathione is a key sulfur-based compound used by bacteria for protection against oxidative stress and by the coral holobiont as a source of organic sulfur [3,35,87]. Taurine oxidation is coupled to sulfur oxidation to increase sulfur availability in the coral holobiont. The degradation of the amino acid taurine produces thiosulfate, which is converted to sulfate via sulfur oxidation [10,88]. Sulfur oxidation genes increase in relative abundance under stress and bleaching [3,58], and here, we show that these genes increase in pre-bleaching conditions. Taurine oxidation by *Ruegeria* was coupled to high primary productivity by planktonic dinoflagellates and plays a key role in the organic sulfur turnover in pelagic environments [89]. Here, *Ruegeria* could be playing a similar role by increasing sulfur availability via taurine oxidation to the dinoflagellates *Symbiodiniaceae* in the coral holobiont under heat exposure.

### 4.5. Stress Response

The proportions of bacterial stress response genes in the coral SML metagenomes were not affected by the heat treatment. Therefore, the coral holobiont was not engaging with the increased temperature with a generalized stress response, as the coral–algal physiology showed relatively mild decreases in GP:R ratios and Fv/Fm. Metagenomics describes the relative abundance of genes in the microbiome and identifies functional genes that are potentially being selected by the microbiome in each condition [10,55,56] but it does not measure which functional genes are being expressed at the point at which the sample was taken. Gene abundance derived from metagenomes and corresponding transcript abundance derived from metatranscriptomes are highly correlated in coral [90] and human gut [91] microbiomes, while some specific gene pathways can be over- or under expressed. Therefore, bacterial stress response gene expression could be increased under heat exposure, but the metagenomes showed that they were not selected by the coral SML microbiome in this experiment. Future studies coupling metagenomics with metatranscriptomics could elucidate the changes in stress response genes in the coral microbiome under increased temperatures.

### 4.6. Connecting Coral Physiology to Microbiome Structure

The photochemical efficiency of the coral endosymbionts (Fv/Fm) can be directly modulated by coral microbiome manipulation [31,32] and is a key physiological factor explaining the coral microbiome structure under heat exposure [92]. Here, Fv/Fm significantly explained the variation in coral–microbial taxonomic and functional profiles (nitrogen and sulfur metabolisms). In contrast, GP:R did not show a statistically significant effect on the *P. strigosa* SML microbiome. GP:R captures the physiological response of both the coral animal and the symbionts, as dark respiration rates (R) are measured for the whole colony. Fv/Fm captures the photosynthetic response of the coral algal endosymbiont. We show a the microbial-level relationship between the *Symbioniaceae* and the SML bacteria, which is more sensitive to external fluctuations than the coral animal physiology [92]. Lower Fv/Fm could explain the changes in bacterial metabolism that were supporting the *Symbioniaceae* challenged by higher temperatures via nitrogen and sulfur acquisition (Figure 6).

## 5. Conclusions

As thermal stress and disease outbreaks are some of the leading causes of an unprecedented loss in coral cover worldwide [19,21,22,93], a deeper understanding of the coral microbiome and its response to temperatures stress enhances our ability to infer how corals react to stress and potentially develop strategies to mitigate damage. We used shotgun metagenomics in an experimental setting to understand the dynamics of microbial taxa and genes in heat-exposed corals.

The metagenomes of corals exposed to heat showed greater similarity when compared to ambient treatments. We hypothesize that this low β-diversity is the result of a selective pressure towards a beneficial microbiome that supports the holobiont in withstanding temperature changes (Figure 6). The microbiome of pre-bleached corals maintained holobiont homeostasis by controlling opportunistic pathogens and providing nitrogen and sulfur to the algal symbiont and energy to the coral host.

The caveats of this study include the tank effect on the coral microbiome and a reduced number of replicates. The hypotheses suggested by our results need to be further explored by expanding the scope to examine other niches of the coral, the way in which the microbiome changes across a time series analysis, and how the microbiome changes as it approaches and surpasses the bleaching threshold.

Promising conservation efforts have been focusing on promoting and maintaining coral microbiome health [94,95,96], including the development of coral probiotics [30,31,32]. Here we support these efforts by showing that heat exposure is leading to potentially beneficial coral microbiomes.

## Figures and Tables

**Figure 1 microorganisms-12-01005-f001:**
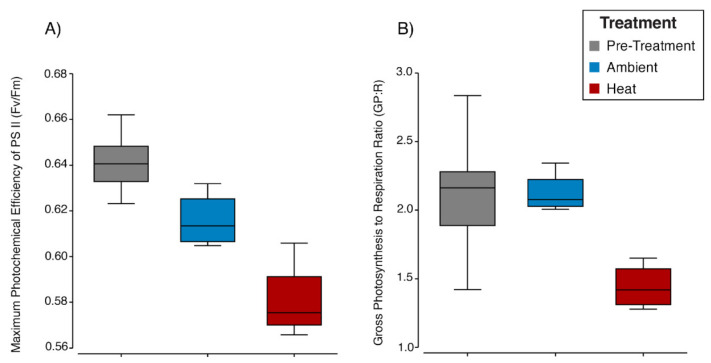
Maximum photochemical efficiency of photosystem II (Fv/Fm) (**A**) and gross productivity to dark respiration ratios (GP:R) (**B**) of coral colonies (total *n* = four per treatment) after exposure to heat and ambient temperatures for one week.

**Figure 2 microorganisms-12-01005-f002:**
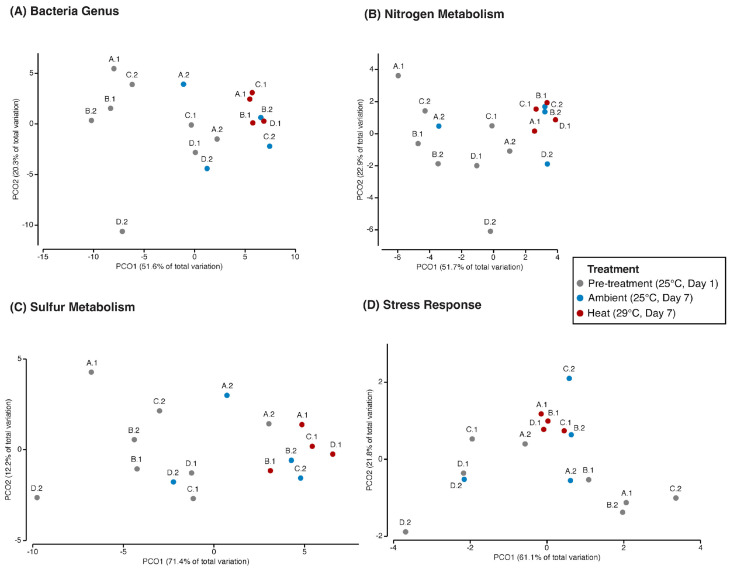
Principal coordinate analyses of the relative abundances of bacterial genera (**A**), nitrogen metabolism (**B**), sulfur metabolism (**C**), and stress response (**D**) genes in the coral SML of *P. strigosa* exposed to different temperature treatments for one week. Four coral colonies (A, B, C, D) were collected from their natural environment and replicated into two fragments (coral nubbins). After an acclimation period of two weeks, the coral nubbins were distributed between heat (A.1, B.1, C.1, D.1) and ambient (A.2, B.2, C.2, D.2) treatments. The analysis was based on a Bray–Curtis similarity matrix.

**Figure 3 microorganisms-12-01005-f003:**
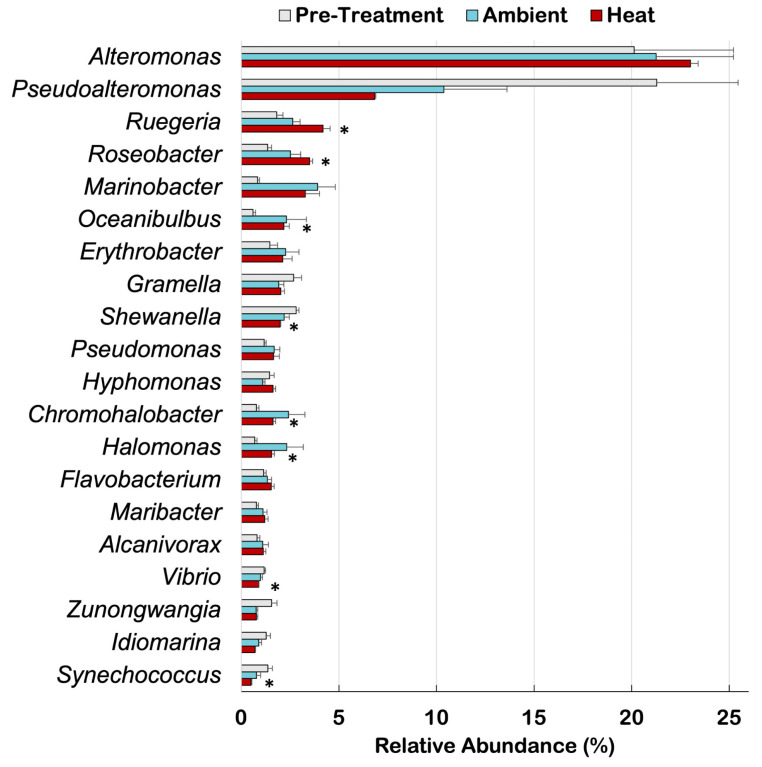
Top 20 most abundant bacterial genera (mean ± SE) across different temperature treatments. Asterisks indicate a significant difference between pre-treatment and heat treatment metagenomes according to Welch’s pairwise comparisons and Benjamini–Hochberg FDR corrections.

**Figure 4 microorganisms-12-01005-f004:**
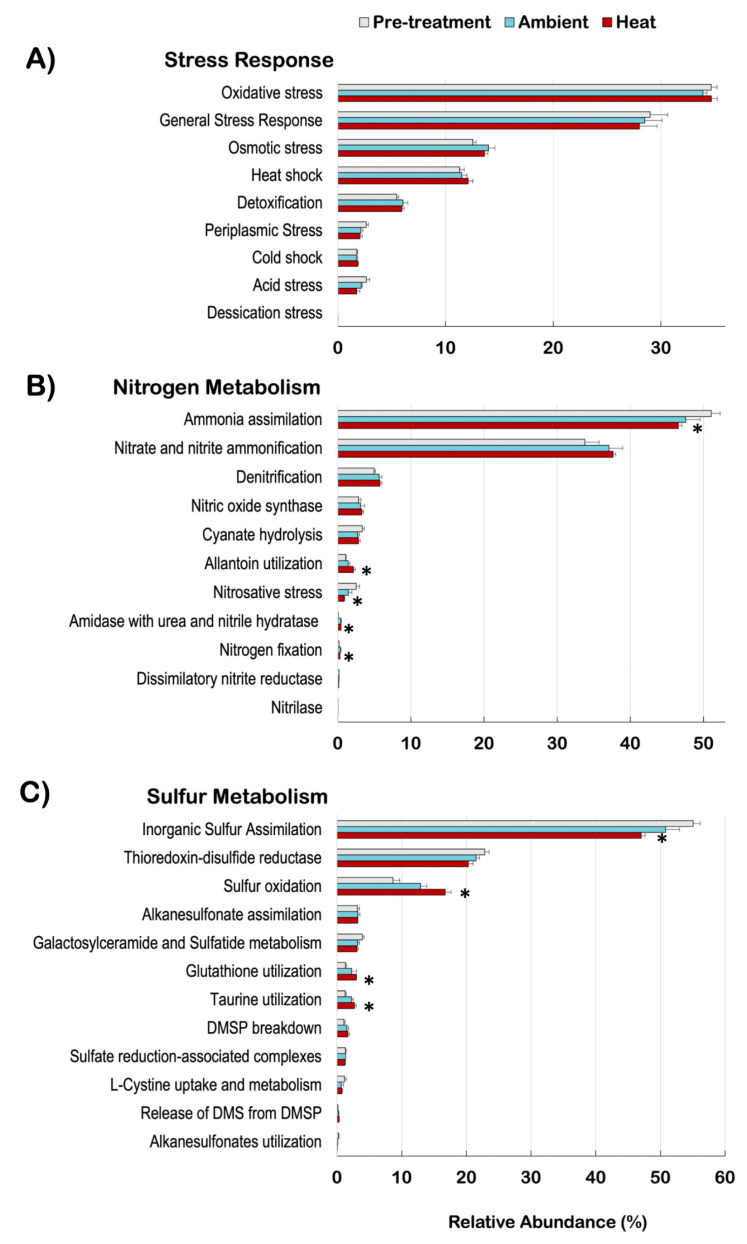
Relative abundances of bacterial genes related to stress response (**A**), nitrogen metabolism (**B**), and sulfur metabolism (**C**). Asterisks indicate a significant difference between pre-treatment and heat-treatment metagenomes according to Welch’s pairwise comparisons and Benjamini–Hochberg FDR corrections.

**Figure 5 microorganisms-12-01005-f005:**
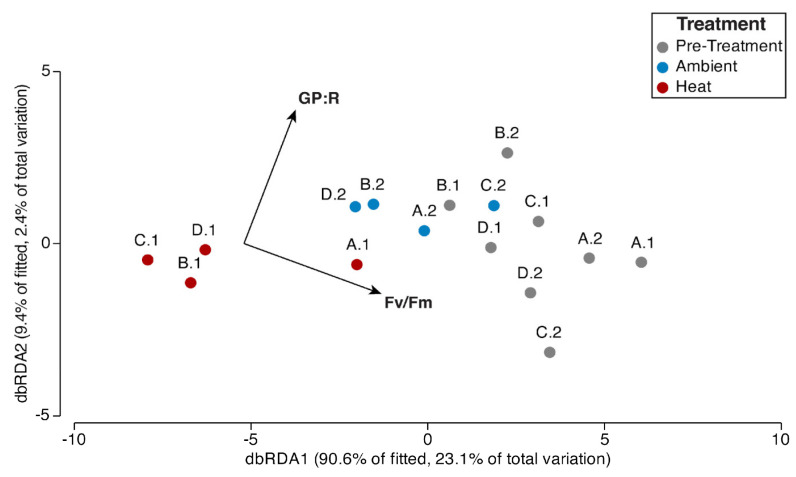
Distance-based redundancy analysis (dbRDA) of the relative abundances of bacterial genera in the coral metagenomes (square root, Bray–Curtis similarity) in the relationship to the coral’s physiological performance (GP:R ratios and Fv/Fm, log (x + 1), Euclidian distances) across temperature treatments). Four coral colonies (A, B, C, D) were collected from their natural environment and replicated into two fragments (coral nubbins). After an acclimation period of two weeks, the coral nubbins were distributed between heat (A.1, B.1, C.1, D.1) and ambient (A.2, B.2, C.2, D.2) treatments.

**Figure 6 microorganisms-12-01005-f006:**
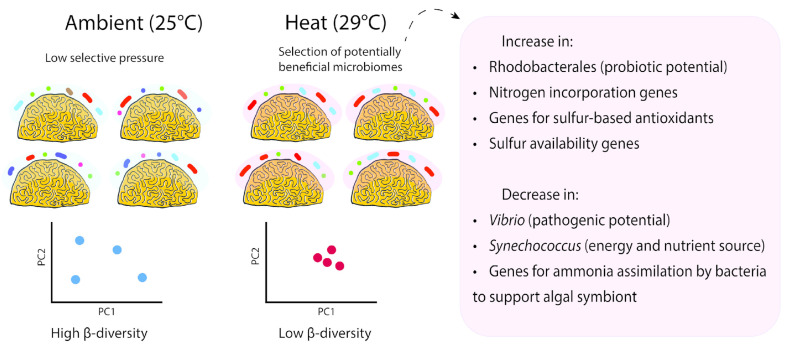
The coral SML microbiome of *P. strigosa* responded to heat exposure with lower β-diversity and a taxonomic and functional composition that was potentially beneficial to the holobiont. Plots are for the visualization of observed patterns and do not represent specific observed data.

## Data Availability

The metagenomic data from this study are publicly available in the SRA database as BioProject PRJNA595374 (https://www.ncbi.nlm.nih.gov/bioproject/595374) and in MG-RAST as public study SDSU_BIOS_2017 (mgp81589; https://www.mg-rast.org/linkin.cgi?project=mgp81589) and (mgp94111; https://www.mg-rast.org/linkin.cgi?project=mgp94111).

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
