# Peer review of "Pre-Bleaching Coral Microbiome Is Enriched in Beneficial Taxa and Functions"

_microorganisms, 2024, doi:10.3390/microorganisms12051005_

Round 1

Reviewer 1 Report

Comments and Suggestions for Authors

Line 20. Define here what SML is.

So you have 2 corals from inner and 2 corals from outer reefs. Each one fragmented to have 8 fragments from the inner and 8 fragments from the outer. You mention only healthy nubbings were used, how many were discarded?

L219. You use two dependent variables Fv/Fm and GP:R with an indepdendent factor with 3 levels (pre treatment, ambient and heat) but present it as a multivariate analysis using permanova. How is this? And if I understand this clearly, you have 4 coral nubbings per treatment, 12 nubbings in total from the original 16? Are these from the inner or outer reefs from which colonies? And although the same nubbings were not submitted to each treatment but they might have the same genetics in different treatments would a repeated measures anova not be a better analysis to take into account the no independence of the replicates?

It is very interesting that the stress of collecting, transporting, fragmenting and acclimating the corals in aquaria seem to have a largest effect that the temperature treatments (Supplementary Figure S1) and line 251 to 262. Why do you imagine a pre-bleaching condition? If the collecting and acclimating stress already caused a large change in the coral microbiome, what are we really observing with the added heat stress? Maybe you could include the in situ data on figure 3 for a better perspective of the changes.

Line 229. After a significant effect due to collection and acclimation on aquaria…

L. 358 or the change was of 25% on average similarity what was the effect size (on average similarity) of the heat treatment if I understand it well is much smaller, right?

The idea that the changes in coral microbiome could be beneficial is tempting but also a little counterintuitive, see for example Pratte and Richardson, 2018 (DAO 131:213-226). In this case physiological responses decrease with the heat treatment and clearly separate the samples in fig 5, I wonder what would happen if you force your coral to bleach and if it recovers.

Author Response

So you have 2 corals from inner and 2 corals from outer reefs. Each one fragmented to have 8 fragments from the inner and 8 fragments from the outer. You mention only healthy nubbings were used, how many were discarded?

No nubbins were discarded – they were all healthy, but we made sure to have extras in case there was an effect from the manipulation.

L219. You use two dependent variables Fv/Fm and GP:R with an indepdendent factor with 3 levels (pre treatment, ambient and heat) but present it as a multivariate analysis using permanova. How is this? 

We chose to run a PERMANOVA because it requires no assumptions about normality and/or homogeneity of variances, it can be applied identically to both univariate and multivariate data, and the test statistic is identical to the conventional F-statistic when calculated using Euclidean distance for a single variable. This information is from the book “Applied Multivariate Statistics in R , 2024, by Jonathan D. Bakker”.

https://uw.pressbooks.pub/appliedmultivariatestatistics/chapter/permanova/

And if I understand this clearly, you have 4 coral nubbings per treatment, 12 nubbings in total from the original 16? Are these from the inner or outer reefs from which colonies? And although the same nubbings were not submitted to each treatment but they might have the same genetics in different treatments would a repeated measures anova not be a better analysis to take into account the no independence of the replicates?

After the acclimation period, eight coral nubbins (A.1, A.2, B.1, B.2, C.1, C.2, D.1, D.2) from four distinct colonies of coral P. strigosa (colonies A and B, from the inner reef zone and colonies C and D from the outer reef zone) were exposed to a high temperature treatment (29°C ± 1°C, nubbins A.1, B.1, C.1, D.1), while the remaining were maintained at ambient conditions (control, 25°C ± 1°C, nubbins A.2, B.2, C.2, D.2). Productivity and respiration rates and photochemical efficiency were measured the day before the heat treatment started (post-acclimation or pre-treatment) and on day 7, which is the final day of the experiment (post-treatment).

We chose to run a PERMANOVA as it does not assume independence of the replicates.

It is very interesting that the stress of collecting, transporting, fragmenting and acclimating the corals in aquaria seem to have a largest effect that the temperature treatments (Supplementary Figure S1) and line 251 to 262. Why do you imagine a pre-bleaching condition? 

Pre-bleaching was assumed because of the reduction in physiological measurements in response to a temperature treatment 4 degrees above ambient for a sustained period of 7 days.

If the collecting and acclimating stress already caused a large change in the coral microbiome, what are we really observing with the added heat stress? Maybe you could include the in situ data on figure 3 for a better perspective of the changes.

We are observing relative changes in the microbiome after a 2-week acclimation where all replicates were exposed to the same conditions. Therefore, the effect of temperature on the microbiome could be evaluated without the confounding factors present in situ: changes in flow, associated benthic community, light, nutrients, etc. 

The stress of collection – aquarium experiment is a well-known effect; we recognize that this is a limitation of laboratory experiments. The coral-associated bacterial community shifts dramatically when placed in aquaria, and requires a 2-week acclimation period to stabilize (Kooperman et al. 2007, Pratte et al. 2015). 

Yet, our results showed that the coral microbiomes at the level of bacterial genera retained a 75% average Bray-Curtis similarity under experimental conditions when compared to in situ conditions (before colony collection). These changes between in situ and experimental data are included in Figure. S1.

Line 229. After a significant effect due to collection and acclimation on aquaria…

We included lines 492-494 to the discussion highlighting to the caveats of this study, in addition to lines 355-360 stating that “As expected, the coral SML microbiome structure changed after coral colonies were removed from their native environment and fragmented. Experimental settings and manipulation inevitably affect the coral microbiome [71, 72], therefore, the results of this experiment may not entirely reflect the microbiome dynamics happening in the natural environment.” 

  1. 358 or the change was of 25% on average similarity what was the effect size (on average similarity) of the heat treatment if I understand it well is much smaller, right?

The average similarity between Pre-Treatment and Heat microbiomes at the level of genus was 86.6%, after a 7-day treatment where only temperature was different. The average similarity between in situ and Pre-Treatment microbiomes at the level of genus was 75%, after a 15-day acclimation period to a laboratory system. 

The idea that the changes in coral microbiome could be beneficial is tempting but also a little counterintuitive, see for example Pratte and Richardson, 2018 (DAO 131:213-226). In this case physiological responses decrease with the heat treatment and clearly separate the samples in fig 5, I wonder what would happen if you force your coral to bleach and if it recovers.

That’s a good point and increasing the temperature until bleaching occur would be a great follow up study. We discussed a potentially beneficial effect, that needs to be further investigated to confirmed. 

Reviewer 2 Report

Comments and Suggestions for Authors

Abstract

Line 15 – can you give some more justification of ‘how’ coral reef health is tightly connected to the coral microbiome – perhaps also consider introducing the term coral holobiont here as well.

Line 15-17 – the second sentence does not logically flow from the first and I’d advise flipping this sentence to mention warming oceans first and the affect this is having on corals and then mention the need for heat tolerance (which will lead into sentence 3). Also consider putting this sentence first, then introduce the importance of the coral microbiome to the coral holobiont.

Line 19 – SML in full at first use.

Introduction

Great first paragraph. However, consider putting para 2 (about coral bleaching etc) first then follow up with this one.

The introduction could do with another paragraph explaining more about what is known about coral microbiomes in terms of their microbial profile and also how the microbes changes between tissue and mucous and why this is important.

Lines 66-70 – can you flesh this out a bit to be more specific about what your hypotheses are.

Methods

Line 77 – be more specific when describing the temperature fluctuations and habitat zones what is a “more fluctuating profile”??

Line 78-80 this sentence may be a bit too speculative, given there is growing evidence of niche differentiation across habitats aligning with genetic differentiation  - can you find a reference to substantiate your statement?

Line 85- how much and how was the mucus collected?

Line 86 – two colonies from two zones sampled is insufficient to describe natural spatial variability.

Lines 110-112 – just a single nubbin of each coral exposed to a single temperature treatment (with the other nubbin per colony used as a control) is a very basic design.

Line 125 – is is not entirely clear why these physiological measurements are needed, that should be established in the hypothesis section of the introduction.

Line 163 – this is the first time its mentioned that mucus was collected from the nubbins prior to the heat treatment. Should come earlier.

Lines 168-197 – need far more detail on the metagenomic workflow incl use of control, technical replication, and the data filtering – atm there is not enough information to comment on the reliability of the results.

Lines 176- pls add references for these databases

Results

Figure 1 – what about the control corals? – are these the ‘ambient’ results?

Line 243 – what about the control nubbins?

Figure 2- there is so much variation between the four corals naturally, with such little biological replication and no technical replication it is tricky to infer any meaningful pattern other than the metagenomes became more similar after the heat treatment.

Figure 3 (& fig 4) -does this data need to be transformed? How come the ambient relative abundance of several genera incl Marinobacter and Oceanibubus is higher than the heat treated ones?

Overall, how come the data is not analyzed at the scale of zone? Corals were sampled from inner and outer reef zones so you need to establish if there are differences related to their zone before pooling.

Disscusion

Line 349-351 – what results are you basing this on?

Line 363-364 – I don’t understand this argument, if the microbiome has ‘simplified’ and become more similar after the temperature stress, how is this beneficial to the holobiont? It is explained more in paragraphs to come but can you provide a bit more info here. Also, the corals were only exposed to quite a moderate level of warming in the experiment, once they approach and surpass their bleaching threshold the microbiome will change again - so be careful not to overinterpret what you have done as it does not adequately represent the thermal stress scenarios corals face on reefs.  Not only because of the moderate warming, but because corals are only exposed for 1DHW - on the reef they are facing much higher DHW. 

lines 384-388 – seems a bit speculative, Synechococcus relative abundance is low across all treatments – so I advise treating the idea that the coral has consumed these bacteria as a hypothesis rather than as fact.

Line 486 – the authors point out the key drawback of this study – limited replication prevents the ability to infer the generality of the results.

Conclusions

Line 491- revise this sentence, the future of coral reefs does not rely on the understanding of the coral microbiome – please consider revising to something like:  A deeper understanding of the coral microbiome and its response to temperatures stress enhances our ability to infer how corals react to stress and potentially develop strategies to mitigate damage.

Coral disease is mentioned in abstract and conclusions but nowhere else, it is not examined in the experiments so seems a bit superfluous to this study.

I quite like figure 6 but I feel the data is very preliminary so this should be cited as more of a 'hypothesis' of how corals respond to 'moderate' levels of temp stress

Lines 502-505 – combine with paragraph above. These results are very preliminary due to a lack of replication so need to say something more about needing to expand the scope of the study and examine other parts of the corals, the influence of niche, the way the microbiome changes across a time-series - and how the microbiome changes at it approaches and surpasses a bleaching threshold. 

I encourage the authors need to consider the potentially negative implications of simplifying the microbiome – a loss of diversity is ‘usually’ risky and negative result – please include a critical review and discussion of the counter argument to the one presented that heat exposure has led to beneficial coral microbiomes. What could the negative consequences of a loss in beta diversity be?

Author Response

Line 15 – can you give some more justification of ‘how’ coral reef health is tightly connected to the coral microbiome – perhaps also consider introducing the term coral holobiont here as well.

Line 15-17 – the second sentence does not logically flow from the first and I’d advise flipping this sentence to mention warming oceans first and the affect this is having on corals and then mention the need for heat tolerance (which will lead into sentence 3). Also consider putting this sentence first, then introduce the importance of the coral microbiome to the coral holobiont.

Yes, we accepted the reviewer’s suggestions and modified lines 15 – 20: “Coral reef health is tightly connected to the coral holobiont, which is the association between the coral animal and a diverse microbiome functioning as a unit. The coral holobiont depends on key services such as nitrogen and sulfur cycling mediated by the associated bacteria. However, these microbial services may be impaired in response to environmental changes, such as thermal stress. A perturbed microbiome may lead to coral bleaching and disease outbreaks, which have caused an unprecedented loss in coral cover worldwide, particularly correlated to a warming ocean.”

Line 19 – SML in full at first use.

We fixed this issue. 

Introduction

Great first paragraph. However, consider putting para 2 (about coral bleaching etc) first then follow up with this one.

The introduction could do with another paragraph explaining more about what is known about coral microbiomes in terms of their microbial profile and also how the microbes changes between tissue and mucous and why this is important.

Lines 66-70 – can you flesh this out a bit to be more specific about what your hypotheses are.

Yes, we accepted the reviewer’ suggestions and modified lines 70 – 81: “(…) Therefore, changes in the proportions of coral-microbial taxa and gene functions in response to heat could reveal key players in the microbiome that may be providing beneficial services to support a thermally challenged holobiont. Here we investigate the effects of increased temperature on the taxonomic and functional profiles of coral SML microbiomes. We addressed these aims by exposing corals to a pre-bleaching heat treatment and analyzing coral-algal physiological parameters and microbial taxa (genus level) and gene functions (nitrogen metabolism, sulfur metabolism, and stress response) associated with the coral SML using shotgun metagenomics. As the coral SML is in intimate contact with the external environment, acting as a conduit between the water column and host physiology, we hypothesize that changes in coral host physiological parameters to be correlated to the dynamics of the SML microbiome.”   

Methods

Line 77 – be more specific when describing the temperature fluctuations and habitat zones what is a “more fluctuating profile”??

Yes, we accepted the reviewer’ suggestions and modified lines 88– 91: “(…)specific thermal regimes ranging from more fluctuating profiles in the inner lagoon patch reefs (~13-15 ºC of temperature difference between winter averages of 16-17 ºC and summer averages of 30-31 ºC) and milder conditions in the outer reefs (~10 ºC of temperature difference between seasonal averages of 19 and 29 ºC)[11, 36–39].”

Line 78-80 this sentence may be a bit too speculative, given there is growing evidence of niche differentiation across habitats aligning with genetic differentiation  - can you find a reference to substantiate your statement?

We could not find any study on the genetic distribution of P. strigosa in Bermuda. We removed the sentence as it does not affect the results or interpretation of our data.

Line 85- how much and how was the mucus collected?

50 ml of mucus diluted in sterile seawater using a custom-made two-way syringe (Line 170-174)

Line 86 – two colonies from two zones sampled is insufficient to describe natural spatial variability.

We agree and that’s why we’re not using reef zone as a factor in our analysis, however, sampling across reef zones is more representative of the reef system and reduces the chances of sampling individuals locally acclimated to the same conditions.

Lines 110-112 – just a single nubbin of each coral exposed to a single temperature treatment (with the other nubbin per colony used as a control) is a very basic design.

We decided to prioritize resources for the DNA sequencing and metagenomic analysis, and the relative low replication was a caveat explicitly included in the conclusion (lines 504-505)

Line 125 – is is not entirely clear why these physiological measurements are needed, that should be established in the hypothesis section of the introduction.

Yes, we accepted the reviewer’ suggestions and modified lines 70 – 81: “(…) As the coral SML is in intimate contact with the external environment, acting as a conduit between the water column and host physiology, we hypothesize that changes in coral host physiological parameters to be correlated to the dynamics of the SML microbiome.”   

Line 163 – this is the first time its mentioned that mucus was collected from the nubbins prior to the heat treatment. Should come earlier.

We believe that the placing of this information is fitting for the “Experimental design” subsection. 

Lines 168-197 – need far more detail on the metagenomic workflow incl use of control, technical replication, and the data filtering – atm there is not enough information to comment on the reliability of the results.

Lines 176- pls add references for these databases

Yes, we accepted the reviewer’ suggestions and modified lines 187 – 193: “The sequenced DNA was analyzed for quality control using PrinSeq before annotation to remove and trim low-quality sequences (i.e., exact duplicates, sequences that contained N’s, and sequences that had a Q-score < 20). The metagenomes were annotated through MG-RAST applying the minimum cutoff parameters of 1 × 10−5 e-value, 70% identity, and alignment length of 30 nucleotides. using the RefSeq database (Pruitt, et al. 2007) for taxonomic annotations and the SEED database (Overbeek, et al. 2014) for functional annotations.”

Results

Figure 1 – what about the control corals? – are these the ‘ambient’ results?

Line 243 – what about the control nubbins?

Correct, the control corals were maintained at the same ambient temperature (reflecting the reef temperature at the time of collection) from the beginning to the end of the acclimation period (pre-treatment) and for the seven-day period while corals in the other treatment were exposed to higher temperatures. 

Figure 2- there is so much variation between the four corals naturally, with such little biological replication and no technical replication it is tricky to infer any meaningful pattern other than the metagenomes became more similar after the heat treatment.

A higher similarity (low beta diversity) and consistency in the microbiome taxonomic and functional profiles under heat is the focus of our discussion and the basis for our interpretation explained in Figure 6.  

Figure 3 (& fig 4) -does this data need to be transformed? How come the ambient relative abundance of several genera incl Marinobacter and Oceanibubus is higher than the heat treated ones?

The asterisks indicate a significant difference between pre-treatment and heat-treatment metagenomes according to Welch’s pairwise comparisons, and Benjamini-Hochberg FDR corrections. No differences between pre-treatment and ambient were detected. Notice the higher error bars in the ambient Marinobacter and Oceanibubus examples. 

Overall, how come the data is not analyzed at the scale of zone? Corals were sampled from inner and outer reef zones so you need to establish if there are differences related to their zone before pooling.

As you mentioned earlier, ‘two colonies from two zones sampled is insufficient to describe natural spatial variability”. That’s why we’re not using reef zone as a factor in our analysis, however, sampling across reef zones is more representative of the reef system and reduces the chances of sampling individuals locally acclimated to the same conditions.

Disscusion

Line 349-351 – what results are you basing this on?

We included “(…) this study, Fig. 2A)”

Line 363-364 – I don’t understand this argument, if the microbiome has ‘simplified’ and become more similar after the temperature stress, how is this beneficial to the holobiont? It is explained more in paragraphs to come but can you provide a bit more info here. Also, the corals were only exposed to quite a moderate level of warming in the experiment, once they approach and surpass their bleaching threshold the microbiome will change again - so be careful not to overinterpret what you have done as it does not adequately represent the thermal stress scenarios corals face on reefs.  Not only because of the moderate warming, but because corals are only exposed for 1DHW - on the reef they are facing much higher DHW. 

An increase of 4C from ambient sustained for 7 days may be not conducive to bleaching in the case of our experiment, but it is significant. For example, similar coral microbiome studies cited in the introduction considered the effects of increased temperatures from ambient 26 °C to 30 °C and from ambient 29 °C to 34 °C heat (for 9 hours) as heat stress. Our discussion focuses on the relative changes seen in an experimental setting, and we did not make any claims that our results are representative of bleaching events in situ. 

lines 384-388 – seems a bit speculative, Synechococcus relative abundance is low across all treatments – so I advise treating the idea that the coral has consumed these bacteria as a hypothesis rather than as fact.

We included “(…) were hypothetically consuming these microbes” (line 404)

Line 486 – the authors point out the key drawback of this study – limited replication prevents the ability to infer the generality of the results.

Yes, this is a caveat of our study.

Conclusions

Line 491- revise this sentence, the future of coral reefs does not rely on the understanding of the coral microbiome – please consider revising to something like:  “A deeper understanding of the coral microbiome and its response to temperatures stress enhances our ability to infer how corals react to stress and potentially develop strategies to mitigate damage.

We included the reviewer’s suggestion in lines 503-506 “As thermal stress and disease outbreaks are some of the leading causes of an unprecedented loss in coral cover worldwide [19, 21, 22, 93], a deeper understanding of the coral microbiome and its response to temperatures stress enhances our ability to infer how corals react to stress and potentially develop strategies to mitigate damage.”

Coral disease is mentioned in abstract and conclusions but nowhere else, it is not examined in the experiments so seems a bit superfluous to this study.

We mention disease outbreaks as an extreme consequence to coral microbiome disruption, bringing a bigger picture to the specificity of our study. 

I quite like figure 6 but I feel the data is very preliminary so this should be cited as more of a 'hypothesis' of how corals respond to 'moderate' levels of temp stress

When we refer to Figure 6 in the Conclusion, we start the sentence with “We hypothesize…” (Line 509). The figure 6 legend highlights that “plots are for visualization of observed patterns and do not represent specific observed data” and we refer to the “potentially beneficial microbes” and not “beneficial microbes” because our interpretation sees a potential, not a confirmed benefit. The rest of the content on the figure was what we observed: increase in Rhodobacterales, nitrogen incorporation genes, etc..; and that is not speculative. 

Lines 502-505 – combine with paragraph above. These results are very preliminary due to a lack of replication so need to say something more about needing to expand the scope of the study and examine other parts of the corals, the influence of niche, the way the microbiome changes across a time-series - and how the microbiome changes at it approaches and surpasses a bleaching threshold. 

We addressed this comment by including lines 515-519 “Caveats of this study include the tank effect on the coral microbiome and a reduced number of replicates. The hypotheses raised from our results need to be further explored in future studies by expanding the scope of this study and examining other parts of the corals, the influence of niche, the way the microbiome changes across a time-series, and how the microbiome changes at it approaches and surpasses a bleaching threshold.” 

I encourage the authors need to consider the potentially negative implications of simplifying the microbiome – a loss of diversity is ‘usually’ risky and negative result – please include a critical review and discussion of the counter argument to the one presented that heat exposure has led to beneficial coral microbiomes. What could the negative consequences of a loss in beta diversity be?

Loss of diversity as in richness is ‘usually’ risky and negative result, lower beta diversity implies greater evenness within that group and similarity of microbial community structure. Greater beta diversity is associated in the literature with microbial dysbiosis. 

Reviewer 3 Report

Comments and Suggestions for Authors

1. Abstract: "SML microbiomes" and "surface mucous layer (SML) microbiomes" should be change.

2. Did you have set parallel groups for A.1, A.2, and B.1, etc.? Is this needed or not?

3. Metagenome can provide many information, why you only focused on stress response, nitrogen and sulfur metabolisms?

4. Vibrios in the Heat group showed a decreasement, why? As far as know, the abundance of Vibrio spp. usually have a positive correlation with temperature.

Author Response

  1. Abstract: "SML microbiomes" and "surface mucous layer (SML) microbiomes" should be change.

We included this change in the abstract.

  1. Did you have set parallel groups for A.1, A.2, and B.1, etc.? Is this needed or not?

Our design included an ambient (control) group and a heat treatment, where the same colony was replicated in each group (colony A: A.1 in one treatment,  A.2 in the other treatment, colony B…).

  1. Metagenome can provide many information, why you only focused on stress response, nitrogen and sulfur metabolisms?

We selected these three broad functional gene groups because they have the greatest relevance for corals under heat stress [2, 3, 17, 57–62], so we could do an in depth analysis of the functions that are most applicable to coral health. (Lines 184 – 193)

  1. Vibrios in the Heat group showed a decreasement, why? As far as know, the abundance of Vibrio spp. usually have a positive correlation with temperature.

We also thought that was an interesting and unusual relationship. We hypothesized that a decrease in Vibrio could be due to an increase in Rhodobacterales, some of which have probiotic potential and can inhibit growth of Vibrio. (Lines 370 - 377)